# Family context and individual characteristics in antenatal care utilization among adolescent childbearing mothers in urban slums in Nigeria

**Akanni Ibukun Akinyemi**[1,2]*, **Temitope Peter Erinfolami**[2], **Samuel Olinapekun Adebayo**[3], **Iqbal Shah**[1], **Reni Elewonbi**[1], **Elizabeth Omoluabi**[3]

1 Department of Global Health & Population, Harvard School of Public Health, Boston, Massachusetts, United States of America, 2 Department of Demography and Social Statistics, Obafemi Awolowo University, Ile Ife, Nigeria, 3 Center for Research Evaluation Resources and Development (CRERD), Ile Ife, Nigeria

* akakanni@yahoo.ca, akakanni2@gmail.com

## Abstract

### Introduction

Adolescent pregnancy contributes significantly to the high maternal mortality in Nigeria. Research evidence from developing countries consistently underscores Antenatal Care (ANC) among childbearing adolescents as important to reducing high maternal mortality. However, more than half of pregnant adolescents in Nigeria do not attend ANC. A major gap in literature is on the influence of family context in pregnant adolescent patronage of ANC services.

### Methods

The study utilized a cross-sectional survey with data collected among adolescent mothers in urban slums in three Nigerian states namely, Kaduna, Lagos, and Oyo. The survey used a multi-stage sampling design. The survey covered a sample of 1,015, 1,009 and 1,088 child-bearing adolescents from each of Kaduna, Lagos, and Oyo states respectively. Data were analyzed at the three levels: univariate, bivariate and multivariate.

### Results

Overall, about 70 percent of female adolescents in our sample compared with 75 percent in the Demographic and Health Survey (DHS) had any antenatal care (ANC) visit. About 62 percent in our sample compared with 70 percent in the DHS had at least 4 ANC visits, and, about 55 percent in our sample compared with 41 percent of the DHS that had 4 ANC visits in a health facility with skilled attendant (4ANC+). Those who have both parents alive and the mother with post-primary education have higher odds of attending 4ANC+ visits. The odds of attending 4ANC+ for those who have lost both parents is almost 60% less than those whose parents are alive, and, about 40% less than those whose mothers are alive.

**Data Availability Statement:** All relevant data are within the paper and Mendeley Data, V1, DOI: 10.17632/mks57dyb5f.1.

**Funding:** The author(s) received no specific funding for this work.

**Competing interests:** The authors have declared that no competing interests exist.

The influence of mother's education on 4ANC+ attendance is more significant with large disparity when both parents are dead.

## Conclusion

The study concludes that identifying the role of parents and community in expanding access to ANC services among adolescent mothers is important in improving maternal health in developing countries.

## Introduction

Adolescent pregnancy contributes significantly to the unacceptably high level of maternal mortality and morbidity in developing countries. Maternal causes account for 15% of female adolescents' death globally, and it is the leading cause of death among adolescent girls aged 15–19 years in Africa [1–3]. The burden is more concentrated in West and Central Africa where 6 per-cent of adolescents reported births before age 15 and 28 per cent of women between the ages of 20 and 24 reported a birth before age 18 [4, 5]. In Nigeria, about a fifth of adolescent girls have begun childbearing and the adolescent fertility rate is 123 births per 1,000 adolescent girls aged 15–19 [5]. Half of these pregnancies are among teenage girls with no education, about 43 per cent of these among the poorest. The consequences of early childbearing especially for unintended pregnancies are reflected in high birth-risk among adolescent mothers [6, 7]. Adolescents who are under 15 years are five times more likely to die during pregnancy or childbirth, and also to be presented with other life challenges [8]. Adolescents have an increased risk of low birth weight, pre-eclampsia/eclampsia, preterm birth, and, maternal and perinatal mortality [1] with high associated birth risk [6].

Antenatal care (ANC) is one of the core interventions for improving maternal outcomes, particularly among young inexperienced women. ANC provides the opportunity for early detection of pregnancy related risks and provision of appropriate treatment. ANC improves women's lives during pregnancy and ensures healthy pregnancy outcomes [9, 10]. It also promotes safe motherhood by identifying and treating actual and potential problems related to pregnancy in a timely manner [11]. Research evidence from developing countries consistently underscores the importance of ANC among childbearing adolescents, most especially in reducing their leading causes of maternal-related deaths such as hemorrhage, sepsis, hypertensive disorders, obstructed labor, and complications of abortion [3, 12, 13].

However, studies have shown that adolescent mothers have lower propensity to attend ANC [9], [10, 14–18], and the attendance is worse off in poor communities [19, 20]. Evidence from Nigeria Demographic and Health Survey showed that only about 35% of adolescent women had at least four ANC visits. In most cases, pregnant adolescents presented for ANC at advanced stages of their pregnancies [21–23]. There is therefore the need to further understand the factors that influence pregnant adolescents' patronage of ANC. This is of significance particularly in addressing the associated risks of adolescent pregnancy and childbearing in resource-constrained societies, and towards attaining SDG3- and SDG5.6.

Evidence from extant literature suggested some of the predictors of ANC attendance among childbearing adolescents to include socio-economic factors [15] including women's education [24–26]. Living in urban areas, and lower birth order were also associated with higher levels of ANC care among young women. Other evidences also suggested that male-partner's education, wealth quintile, and region of residence were important factors associated

with maternal healthcare service utilization [27]. The status of pregnancy, especially unplanned [23], and, partners' factors were linked to the uptake of maternal health care services including ANC [28]. Non-use of ANC among adolescents was linked with social stigma and shame [29].

A major gap in literature is on the influence of the family context in pregnant adolescent patronage of ANC services. Although, family context has been identified as a major predictor of adolescents' sexual and health seeking behavior and family planning [30, 31], there is lack of evidence on the influence of this on ANC during pregnancy. Studies have shown that family context is very important for adolescent prevention of repeat pregnancy [32], exposure to teenage pregnancy [33], and adolescents' parenting knowledge [34]. Besides, there is also a growing concern about urban slum health challenges [35–37]. Evidence suggests that urban slums are characterized by poor maternal care services [38], including among childbearing adolescents in slum areas of developing countries [39, 40]. Studies from urban slum areas in Nigeria have found that they are characterized by poor health services [41] and poor sexual and reproductive health outcomes among women [42]. The study is therefore aimed at examining the influence of family context in ANC uptake among childbearing adolescents in urban slums in Nigeria.

## Methods

Data were collected among childbearing adolescent females in urban slums in three Nigerian states namely Kaduna in the northwest region, and Lagos and Oyo in the southwest. For this study, we have adopted UN-HABITAT's definition of a slum as an urban area characterized with lack of durable facilities, poor access to adequate sanitation and poor living conditions. The study documented the slum areas in the study states with high population of adolescent using population counts from the Nigeria's Geo-Referenced Infrastructure and Demographic Data for Development (GRID3) programme.

The survey used a multi-stage sampling design. At the first stage, the 3 states identified in extant literature as accommodating most urban slums in Nigeria- Kaduna, Lagos and Oyo states were selected for the northern and southern region. All the local government areas (LGAs) where the slums areas were located were selected as the second stage. The GRID3 population estimates shows the slum areas with high population of adolescents and young people and those areas were selected as the study clusters in the urban centers. Households with female adolescents were randomly selected using a referral system until the cluster sample size is attained. A female adolescent was randomly selected in households with more than one eligible female respondent to ensure community representativeness.

As such, individuals were nested within clusters and clusters within LGAs. The data covered a sample of 1,015, 1,009 and 1,088 childbearing adolescents with at least a child less than 5 years from Kaduna, Lagos, and Oyo states respectively. The data collection was implemented between July and October 2018.

The inclusion criteria included being a teenager and having begun childbearing; and having had a child during adolescence with the reference child below 5 years. The upper age limit for those who had a child during adolescence was therefore extended to 24 years to accommodate those who had a child at age 19 years with the current age of the child around 5 years. All women who had their first birth after age 19 years were excluded from the sample.

### Ethics approval and consent to participate

The National Health Research Ethics Committee of Nigeria (NHREC) approved this study and the protocol on May 25, 2018, with IRB number NHREC/01/01/2007. A letter of approval for the publication was granted by Harvard T.H. Chan School of Public Health, IRB18-1385

August 27, 2018. All participants signed written consent form during data collection. In addition, guardians of minors, most especially, the unmarried ones signed assent form.

## Measurement of variables

The outcome variable is measured in terms of whether or not an adolescent made at least 4 ante-natal care visits in facilities with a qualified skilled attendant during the last pregnancy [12, 27, 43]. Though the World Health Organization (2016) has recommended a minimum of 8 antenatal care visits for a positive pregnancy experience, the cutoff of 4 ANC visits in this study was based on what officially obtains in-country at the time of data collection. The 8-contact cut-off was recently officially implemented in Nigeria with the revision of the National Health Management Information System (NHMIS) tools in 2019. The variable was dichotomized as "1" for those who had at least 4 ANC visit in facilities with a qualified skilled attendant and "0" otherwise. The main explanatory variable is the family context measured in terms of parental living status and other family characteristics. Some other explanatory variables already identified in the literature [9, 15, 17, 23, 26, 44] were also included. We added a concept of social status to measure some important individual-level variables that can predispose respondents to more social protection opportunities in their communities, such as owning a personal bank account, national identity card, international passport, membership of a trade union etc. Possession of each of these things was scored "1" and otherwise scored "0". The addition of all the dichotomized variables was then computed as a final score of social status. The final score was however further dichotomized into "1" for respondents with above-median score of social status and "0" otherwise.

## Analysis

Data were analyzed at three levels: univariate, bivariate and multivariate. At the bivariate level, chi-square analyses were performed to examine the association that exists between each of the outcome variables and the explanatory variables. At the multivariate level however, we utilized binary logistic regressions to examine the effects of the explanatory variables on the healthcare utilization variables.

## Results

The background characteristics of the respondents as presented in Table 1 show that about one-tenth of the respondents were 16 years or less, about 60 percent were aged 18–19 years while 22% were 20 years or more. About 77 percent had secondary education, 34 percent are currently working. About 69 percent are currently in marital union while about 44 percent are below medium wealth quintiles. More than half of the respondents were of low social status according to our definition. About 16% had sexual debut by age 15 years or below, and mostly in age-mix relationships with partners 5 or more years older. Only about one-third of these young women had the intention to get pregnant at the time the pregnancy occurred. More than half (57%) of these young mothers had both parents alive, while about 14 percent had both parents' dead. About 55 percent of their mothers had post-primary education, 35 percent of their parents were in a polygamous family, 16 percent of either parent had begun childbearing at teen age while about 6 percent had other siblings who are also parents at teen age.

The outcome variable is as presented in Table 2 below. Overall, in the three states, about 70 percent had any ANC visit, about 62 percent had at least 4 ANC visits, and about 55 percent had 4 ANC in a health facility with skilled attendant. Across the states, lower proportions of young mothers in Kaduna had any ANC (58%), and 4 ANC (43%), compared with Oyo and Lagos.

**Table 1. Percentage distribution of respondents, by background characteristics.**

|  | No. | % |
|---|---|---|
| State |  |  |
| Kaduna | 1,015 | 32.6 |
| Lagos | 1,009 | 32.4 |
| Oyo | 1,088 | 35.0 |
| Age |  |  |
| 14 | 49 | 1.6 |
| 15 | 94 | 3.0 |
| 16 | 186 | 6.0 |
| 17 | 349 | 11.2 |
| 18 | 916 | 29.4 |
| 19 | 829 | 26.6 |
| 20+ | 689 | 22.1 |
| Education |  |  |
| None | 200 | 6.4 |
| Primary | 384 | 12.3 |
| Secondary | 2,403 | 77.2 |
| Tertiary | 125 | 4.0 |
| Working | 1,051 | 33.8 |
| Muslim | 1,926 | 61.9 |
| Christian | 1,179 | 37.9 |
| Marital Status |  |  |
| Single | 882 | 28.3 |
| Married | 2,141 | 68.8 |
| Other | 89 | 2.9 |
| Wealth status |  |  |
| Poorest | 942 | 30.3 |
| Poorer | 424 | 13.6 |
| Middle | 765 | 24.6 |
| Richer | 469 | 15.1 |
| Richest | 512 | 16.5 |
| Social Status |  |  |
| Low | 1,700 | 54.6 |
| Middle | 730 | 23.5 |
| High | 682 | 21.9 |
| Age at first sex < 15 | 488 | 15.7 |
| Partner 5 years older | 2,926 | 94.0 |
| Intention to get pregnant | 1046 | 33.61 |
| Parents alive? |  |  |
| Only father Alive | 253 | 8.1 |
| Only mother Alive | 643 | 20.7 |
| Both Alive | 1,788 | 57.5 |
| Both Dead | 428 | 13.8 |
| Father has post primary education | 2,172 | 69.8 |
| Mother has post primary education | 1,720 | 55.3 |
| Polygamous background | 1,101 | 35.4 |
| Either parent had children at teen age | 485 | 15.6 |
| Any sibling had a child at teen age | 178 | 5.7 |

**Table 2. Percentage of adolescent mothers having at least 4 ANC visits.**

|  | Adolescent Survey 2018 | |
| --- | --- | --- |
|  | N | % |
| **Any ANC visit** |  |  |
| Kaduna | 1015 | 58.0 |
| Lagos | 1009 | 75.6 |
| Oyo | 1088 | 75.6 |
| **Total** | **3112** | **69.7** |
| **4 ANC visits** |  |  |
| Kaduna | 1015 | 43.4 |
| Lagos | 1009 | 72.1 |
| Oyo | 1088 | 69.0 |
| **Total** | **3112** | **61.6** |
| **4 or more with Skilled ANC care** |  |  |
| Kaduna | 1015 | 43.3 |
| Lagos | 1009 | 58.4 |
| Oyo | 1088 | 62.5 |
| **Total** | **3112** | **54.9** |

We examined at the bivariate level the factors that may significantly influence young mothers to have any ANC visit, 4 or more ANC (ANC) visits and 4 or more ANC (4ANC) visits in facility with a skilled provider (4ANC+) by background characteristics in Table 3 and by family variable in Table 4. The proportion of adolescent and young women in Kaduna who had 4ANC/4ANC+ was lower (43%) compared with other states. However, there was no consistency in the proportion of young women who had 4ANC/4ANC+ with age. Those with higher education and currently engaged in economic activity, married, and of higher social status have higher proportions than others to have 4ANC/4ANC+. All the background variables of interest were statistically significant ($p < .05$) except religion, those with partners of 5 years or more and intention to be pregnant. Across the family variables, those whose mothers are alive or who have both parents alive had a higher proportion of attending 4ANC/4ANC+ with lower proportion when both parents are dead. Across the family variables, parental education, or earlier exposure to childbearing among parents or siblings were not significant predictors of adolescent mothers' patronage of 4ANC/4ANC+.

Results of the adjusted logistic regression models for the determinants 4ANC+ are presented in Table 5. Model 1 shows the influence of respondents' background characteristics on 4ANC+, model 2 shows the isolated influence of the family variables on 4ANC+ while the third model shows the combined effect of both the background and family variables. The results for the influence of background variable for both models 1 and 3 are consistent. While age, religion, and age at first sex were not significant predictors of 4ANC+, location was especially important as young mothers in urban slums in Lagos and Oyo States were almost 3 times as likely as their counterparts in Kaduna to have 4ANC+. Similarly, education was crucial, as women with secondary education compared with their uneducated counterpart were almost 3 times more likely to report 4ANC+. Marriage and social status were also important as married young mothers and those of high social status are twice more likely than the unmarried and of lower social status to have 4ANC+. Furthermore, young mothers who had the intention to be pregnant at the time of the pregnancy were twice more likely than those who considered the pregnancy as unintended to attend 4ANC+. Models 2 and 3 also show consistent results on the influence of family context on 4ANC+. Isolating for parents' life status and

**Table 3. Bivariate analysis between individual variables and outcome variables.**

| | | Any ANC visit | 4 or more ANC visits | 4 or more ANC visits at health facility |
|---|---|---|---|---|
| | **No.** | **%** | **%** | **%** |
| State | | | | |
| Kaduna | 1,015 | 58.0 | 43.3 | 43.3 |
| Lagos | 1,009 | 75.6 | 72.1 | 58.4 |
| Oyo | 1,088 | 75.6 | 69.0 | 62.5 |
| | | $X^2 = 100.1$ P = 0.000 | $X^2 = 214.9$ P = 0.000 | $X^2 = 85.0$ P = 0.000 |
| Age | | | | |
| 14 | 49 | 69.4 | 65.3 | 61.2 |
| 15 | 94 | 55.3 | 54.3 | 47.9 |
| 16 | 186 | 72.0 | 65.1 | 59.7 |
| 17 | 349 | 69.3 | 61.3 | 51.9 |
| 18 | 916 | 72.8 | 66.0 | 58.5 |
| 19 | 829 | 73.7 | 64.5 | 57.8 |
| 20+ | 689 | 63.0 | 52.2 | 47.5 |
| | | $X^2 = 35.0$ P = 0.000 | $X^2 = 39.5$ P = 0.000 | $X^2 = 28.7$ P = 0.000 |
| Education | | | | |
| None | 200 | 47.0 | 41.0 | 36.0 |
| Primary | 384 | 60.9 | 51.8 | 45.3 |
| Secondary | 2,403 | 72.4 | 64.0 | 57.1 |
| Tertiary | 125 | 84.8 | 79.2 | 73.6 |
| | | $X^2 = 84.8$ P = 0.000 | $X^2 = 73.7$ P = 0.000 | $X^2 = 65.3$ P = 0.000 |
| Employment | | | | |
| Not Working | 2,061 | 67.1 | 58.3 | 52.9 |
| Working | 1,051 | 75.4 | 68.1 | 58.9 |
| | | $X^2 = 22.8$ P = 0.000 | $X^2 = 28.3$ P = 0.000 | $X^2 = 10.2$ P = 0.001 |
| Religion | | | | |
| Christianity | 1,179 | 71.3 | 64.1 | 55.0 |
| Islam | 1,926 | 69.1 | 60.2 | 54.9 |
| Traditional religion | 7 | 42.9 | 28.6 | 28.6 |
| | | $X^2 = 4.2$ P = 0.121 | $X^2 = 7.9$ P = 0.019 | $X^2 = 2.0$ P = 0.373 |
| Marital Status | | | | |
| Single | 882 | 60.9 | 53.9 | 44.2 |
| married | 2,141 | 73.8 | 65.1 | 59.4 |
| Others | 89 | 62.9 | 55.1 | 53.9 |
| | | $X^2 = 51.9$ P = 0.000 | $X^2 = 35.1$ P = 0.000 | $X^2 = 57.9$ P = 0.000 |
| Wealth Status | | | | |
| Poorest | 942 | 64.3 | 53.9 | 48.4 |
| Poorer | 424 | 71.7 | 63.7 | 58.0 |
| Middle | 765 | 74.9 | 66.9 | 58.6 |
| Richer | 469 | 72.5 | 64.4 | 58.2 |
| Richest | 512 | 68.6 | 63.7 | 55.9 |
| | | $X^2 = 25.6$ P = 0.000 | $X^2 = 35.9$ P = 0.000 | $X^2 = 24.1$ P = 0.000 |
| Social Status | | | | |
| Low | 1,700 | 64.1 | 55.6 | 49.1 |
| Middle | 730 | 74.4 | 65.6 | 59.0 |
| High | 682 | 79.3 | 72.4 | 65.0 |
| | | $X^2 = 62.7$ P = 0.000 | $X^2 = 64.8$ P = 0.000 | $X^2 = 55.9$ P = 0.000 |

(*Continued*)

**Table 3.** (Continued)

| | No. | Any ANC visit | 4 or more ANC visits | 4 or more ANC visits at health facility |
|---|---|---|---|---|
| | | % | % | % |
| Age at 1st sex | | | | |
| < 14 | 2,624 | 69.2 | 60.4 | 54.0 |
| 15+ | 488 | 73.2 | 68.2 | 60.0 |
| | | $X^2 = 3.0$ P = 0.084 | $X^2 = 10.7$ P = 0.001 | $X^2 = 6.1$ P = 0.013 |
| Partner not 5 years older | 186 | 70.4 | 65.1 | 56.5 |
| Partner 5 years older | 2,926 | 69.8 | 61.4 | 54.8 |
| | | $X^2 = 0.03$ P = 0.861 | $X^2 = 1.0$ P = 0.322 | $X^2 = 0.2$ P = 0.664 |
| Did not intend to get pregnant | 1,046 | 69.3 | 62.5 | 54.4 |
| Intended to get pregnant | 2,066 | 70.9 | 59.8 | 56.0 |
| | | $X^2 = 0.87$ P = 0.35 | $X^2 = 2.12$ P = 0.15 | $X^2 = 0.78$ P = 0.38 |

other family factors in model 2, analysis shows that the demise of one or both parents reduced the odds of 4ANC+. However, having mothers with a post primary education and coming from a polygamous background significantly increased 4ANC+, though these were no longer significant in model 3. Model 3 further reiterates the importance of parents' life status as having both parents dead significantly reduced the odds of attending 4ANC+ by about 40%. However, many of the background variables remain significant after the introduction of family variables.

**Table 4. Bivariate analysis between family variables and outcome variables.**

| | No. | Any ANC visit | 4 or more ANC visits | 4 or more ANC visits at health facility |
|---|---|---|---|---|
| | | % | % | % |
| Parents alive? | | | | |
| Only father Alive | 253 | 68.0 | 60.9 | 54.9 |
| Only mother Alive | 643 | 68.0 | 59.9 | 54.0 |
| Both Alive | 1,788 | 74.3 | 65.5 | 58.0 |
| Both Dead | 428 | 55.4 | 48.4 | 43.5 |
| | | $X^2 = 60.7$ P = 0.000 | $X^2 = 44.4$ P = 0.000 | $X^2 = 29.8$ P = 0.000 |
| Father has no post-primary education | 940 | 70.4 | 60.0 | 54.4 |
| Father has post-primary education | 2,172 | 69.6 | 62.3 | 55.2 |
| | | $X^2 = 0.2$ P = 0.650 | $X^2 = 1.5$ P = 0.218 | $X^2 = 0.2$ P = 0.682 |
| Mother has no post-primary education | 1,392 | 68.7 | 59.0 | 53.8 |
| Mother has post-primary education | 1,720 | 70.8 | 63.8 | 55.8 |
| | | $X^2 = 1.7$ P = 0.197 | $X^2 = 7.5$ P = 0.006 | $X^2 = 1.3$ P = 0.263 |
| Monogamous background | 2,011 | 68.4 | 60.2 | 53.3 |
| Polygamous background | 1,101 | 72.6 | 64.2 | 57.9 |
| | | $X^2 = 6.0$ P = 0.015 | $X^2 = 4.8$ P = 0.028 | $X^2 = 6.3$ P = 0.012 |
| Neither parent had children at teen age | 2,627 | 69.1 | 61.2 | 54.7 |
| Either parent had children at teen age | 485 | 74.0 | 64.1 | 56.1 |
| | | $X^2 = 4.7$ P = 0.030 | $X^2 = 1.5$ P = 0.219 | $X^2 = 0.3$ P = 0.574 |
| No sibling had a child at teen age | 2,934 | 69.8 | 61.5 | 55.0 |
| Any sibling had a child at teen age | 178 | 70.8 | 63.5 | 53.9 |
| | | $X^2 = 0.1$ P = 0.781 | $X^2 = 0.3$ P = 0.601 | $X^2 = 0.1$ P = 0.786 |

**Table 5. Binary logistic regression analysis showing the effect of background and family variables on making at least 4 antenatal visits in facilities with skilled attendant.**

| VARIABLES | OR | C.I. | OR | C.I. | OR | C.I. |
|---|---|---|---|---|---|---|
| **State [RC = Kaduna]** | | Model 1*** | | Model 2*** | | Model 3*** |
| Lagos | 2.33*** | [1.88, 2.90] | | | 2.27*** | [1.82, 2.83] |
| Oyo | 2.57*** | [2.08, 3.17] | | | 2.41*** | [1.94, 3.00] |
| Age [RC = 14] | | | | | | |
| 15 | 0.68 | [0.32, 1.45] | | | 0.69 | [0.32, 1.46] |
| 16 | 1.09 | [0.55, 2.17] | | | 1.09 | [0.54, 2.19] |
| 17 | 0.78 | [0.40, 1.51] | | | 0.76 | [0.39, 1.48] |
| 18 | 0.98 | [0.51, 1.89] | | | 0.97 | [0.50, 1.87] |
| 19 | 0.95 | [0.49, 1.85] | | | 0.95 | [0.49, 1.85] |
| 20+ | 0.59 | [0.30, 1.15] | | | 0.6 | [0.31, 1.18] |
| Education [RC = None] | | | | | | |
| Primary | 1.33 | [0.92, 1.92] | | | 1.31 | [0.90, 1.89] |
| Secondary | 2.07*** | [1.49, 2.86] | | | 1.94*** | [1.40, 2.69] |
| Higher | 3.65*** | [2.15, 6.18] | | | 3.31*** | [1.94, 5.63] |
| Working | 1.09 | [0.92, 1.28] | | | 1.07 | [0.91, 1.27] |
| Religion [RC = Christians] | | | | | | |
| Muslims | 0.91 | [0.76, 1.09] | | | 0.91 | [0.76, 1.09] |
| Traditionalists | 0.34 | [0.06, 1.92] | | | 0.33 | [0.06, 1.89] |
| Marital Status [RC = Single] | | | | | | |
| Married | 1.94*** | [1.60, 2.34] | | | 2.01*** | [1.66, 2.43] |
| Other | 2.37*** | [1.47, 3.82] | | | 2.40*** | [1.48, 3.87] |
| Wealth Status [RC = Poorest] | | | | | | |
| Poorer | 1.37* | [1.08, 1.75] | | | 1.36* | [1.06, 1.73] |
| Middle | 1.29* | [1.05, 1.58] | | | 1.27* | [1.03, 1.56] |
| Richer | 1.23 | [0.96, 1.58] | | | 1.19 | [0.93, 1.53] |
| Richest | 1.1 | [0.85, 1.42] | | | 1.05 | [0.81, 1.36] |
| Social status [RC = Low] | | | | | | |
| Middle | 1.28* | [1.05, 1.54] | | | 1.28* | [1.06, 1.56] |
| High | 1.62*** | [1.30, 2.01] | | | 1.62*** | [1.30, 2.02] |
| Age at first sex > 17 | 1.22 | [0.96, 1.56] | | | 1.23 | [0.96, 1.58] |
| Partner 5 years older | 1.15 | [0.84, 1.57] | | | 1.16 | [0.84, 1.59] |
| Intended to get pregnant | 1.36** | [1.12, 1.67] | | | 1.36** | [1.11, 1.66] |
| Parents status [RC = Both Parents Alive] | | | | | | |
| Only father Alive | | | 0.75* | [0.56, 1.00] | 0.92 | [0.67, 1.25] |
| Only mother Alive | | | 0.83 | [0.68, 1.03] | 1.01 | [0.81, 1.26] |
| Both Dead | | | 0.46*** | [0.36, 0.59] | 0.60*** | [0.46, 0.78] |
| Father has post primary education | | | 1.15 | [0.95, 1.40] | 1.13 | [0.92, 1.40] |
| Mother has post primary education | | | 1.29** | [1.07, 1.55] | 1.14 | [0.93, 1.39] |
| Polygamous background | | | 1.23* | [1.04, 1.44] | 1.15 | [0.97, 1.37] |
| Either parent had child as teenager | | | 0.94 | [0.76, 1.16] | 0.91 | [0.73, 1.13] |
| Any sibling had child as teenager | | | 0.96 | [0.70, 1.30] | 0.95 | [0.68, 1.31] |
| **Constant** | **0.15***** | **[0.07, 0.34]** | **1.07** | **[0.91, 1.25]** | **0.15***** | **[0.06, 0.34]** |

*** p<0.001

** p<0.01

* p<0.05

OR Odds Ratio, C.I Confidence Interval, RC Reference Category.

Table 6 below presents the result of heterogeneity test and the effect of parental factors on 4ANC+ attendance for young pregnant women. The likelihood of attending 4ANC+ among young pregnant mothers in Kaduna state was much lower compared with other states when both parents are dead than when both are alive. The influence of education on 4ANC+ attendance was more significant with a larger disparity when both parents are dead. Those who had post-secondary education were nine times more likely than those without any education to report 4ANC+ among those whose parents were dead. The importance of a polygamous background is only significant when mother is alive, while wantedness of pregnancy was only significant when both parents are alive and being from a polygamous family was significant for those whose mothers are alive.

## Discussion

The study provides insight into maternal health issues among a sub-population of vulnerable women in Nigeria. This study further shows that access to reproductive health services is an important consideration to improve health outcomes among pregnant adolescents (Patra, 2016). Findings from this study shows that about half of childbearing adolescent girls in urban slums did not attend 4ANC+. Thus, constituting a major risk-sustaining factor for high-risk births, which further portend greater incidence of maternal morbidity and mortality. The choices and opportunities adolescents childbearing mothers have in terms of access to sexual and reproductive health information and services will significantly affect the burden of diseases and nations' human capital [2].

Our findings further corroborate other evidence that socio-economic status of women is an important determinant of maternal health utilization during pregnancy [15, 23, 26]. For instance, women's social status as well as education significantly showed the highest disparity, even in the cases where both parents are not alive. This reinforces the universally positive effect of education in reproductive health decision making irrespective of location [45, 46]. On the contrary, however, increasing wealth status in this study reduced 4ANC+ and was not a significant determinant of 4ANC+ in the higher wealth groups. This implies that while improving wealth is helpful to improve ANC utilization and maternal health, it might require a longer time to achieve in resource-poor settings. The study further shows that adolescents with unintended pregnancy are less likely to utilize ANC services, controlling for other factors. This might indicate a fear of stigmatization in their immediate environment. Also, considering that this was most significant when both parents are alive further suggests the influence of parent in making the pregnancy of their adolescent daughter a secret for the avoidance of societal shame that is associated with adolescent pregnancy, most especially in the South [47]. Furthermore, increased antenatal care utilization among the ever-married women suggest the effect of support system and previous pregnancy experience.

Finally, our findings show the survival of parents as a significant determinant of maternal health service utilization for childbearing adolescents. This is important within the African context as parents and family make up social capital and support network for young people. Previous studies have identified parents as important determinants in sexual and reproductive health outcomes of adolescents [30, 48] particularly during pregnancy [33]. Specifically, in northern Nigeria, we found that the effect of having a mother alive had significant implication for young women's patronage of 4ANC+. One possible explanation for this is the fact that mothers are likely to provide adequate informational guidance to their daughters based on their own personal experiences of childrearing and birth preparedness. For adolescents from polygamous families, the influence of the mothers of adolescent mothers may be very significant, suggesting a level of competitiveness among co-wives to provide the utmost support for

**Table 6. Binary logistic regression analysis showing the effect of background and family variables on making at least 4 antenatal in facilities with skilled attendant, controlling for parents' status.**

| VARIABLES | Both Parent Alive OR | C.I. | Only father Alive OR | C.I. | Only mother Alive OR | C.I. | Both parents Dead OR | C.I. |
|---|---|---|---|---|---|---|---|---|
| State [RC = Kaduna] | Model 4*** | | Model 5** | | Model 6*** | | Model 7*** | |
| Lagos | 2.27*** | [1.67, 3.08] | 5.41*** | [2.26, 12.96] | 1.61* | [1.01–2.57] | 3.65*** | [1.83–7.27] |
| Oyo | 2.23*** | [1.67, 2.98] | 6.54*** | [2.65, 16.13] | 2.44*** | [1.49–3.97] | 3.60*** | [1.81–7.16] |
| Age [RC = 14] | | | | | | | | |
| 15 | 0.97 | [0.32, 2.96] | 2.13 | [0.20, 22.37] | 0.22 | [0.03–1.54] | 0.11 | [0.01–1.19] |
| 16 | 1.15 | [0.40, 3.31] | 6.47 | [0.81, 51.55] | 0.36 | [0.06–2.06] | 3.02 | [0.51–17.82] |
| 17 | 0.95 | [0.34, 2.64] | 1.92 | [0.28, 13.29] | 0.23 | [0.04–1.22] | 0.71 | [0.12–4.11] |
| 18 | 1.05 | [0.38, 2.86] | 3 | [0.45, 20.08] | 0.4 | [0.07–2.13] | 1.89 | [0.36–9.88] |
| 19 | 1.13 | [0.41, 3.11] | 2.07 | [0.29, 14.82] | 0.36 | [0.07–1.94] | 1.31 | [0.23–7.28] |
| 20+ | 0.56 | [0.20, 1.56] | 3.89 | [0.56, 27.14] | 0.3 | [0.05–1.63] | 0.82 | [0.15–4.50] |
| Education [RC = None] | | | | | | | | |
| Primary | 1.06 | [0.62, 1.80] | 0.48 | [0.09, 2.58] | 1.17 | [0.53–2.59] | 2.59* | [1.03–6.50] |
| Secondary | 1.59 | [1.00, 2.52] | 0.79 | [0.17, 3.58] | 2.13* | [1.06–4.28] | 3.13** | [1.38–7.15] |
| Higher | 2.22* | [1.08, 4.56] | 1.67 | [0.19, 14.46] | 6.38** | [1.84–22.16] | 9.37** | [2.03–43.31] |
| Working | 1.02 | [0.82, 1.27] | 1.13 | [0.58, 2.21] | 1.31 | [0.90–1.89] | 0.9 | [0.54–1.50] |
| Religion [RC = Christians] | | | | | | | | |
| Muslims | 0.82 | [0.65, 1.05] | 1.39 | [0.67, 2.86] | 0.84 | [0.55–1.27] | 1.3 | [0.72–2.35] |
| Traditionalists | 0.44 | [0.06, 2.99] | | | | | | |
| Marital Status [RC = Single] | | | | | | | | |
| Married | 1.84*** | [1.43, 2.36] | 3.75** | [1.69, 8.35] | 2.45*** | [1.59–3.78] | 2.07* | [1.11–3.86] |
| Other | 1.75 | [0.87, 3.49] | 1.59 | [0.20, 12.97] | 4.74** | [1.84–12.20] | 1.57 | [0.42–5.84] |
| Wealth Status [RC = Poorest] | | | | | | | | |
| Poorer | 1.41* | [1.02, 1.95] | 1.07 | [0.42, 2.69] | 1.19 | [0.66–2.16] | 1.36 | [0.70–2.65] |
| Middle | 1.34* | [1.01, 1.76] | 0.6 | [0.27, 1.30] | 0.97 | [0.60–1.58] | 2.40** | [1.32–4.37] |
| Richer | 1.35 | [0.98, 1.87] | 0.58 | [0.21, 1.57] | 1.18 | [0.66–2.12] | 0.77 | [0.36–1.65] |
| Richest | 1.15 | [0.81, 1.62] | 0.98 | [0.35, 2.78] | 0.87 | [0.50–1.53] | 1.17 | [0.55–2.50] |
| Social status [RC = Low] | | | | | | | | |
| Middle | 1.30* | [1.01, 1.67] | 2.03 | [0.92, 4.48] | 1.36 | [0.86–2.15] | 1.23 | [0.71–2.13] |
| High | 1.78*** | [1.33, 2.38] | 0.81 | [0.33, 2.01] | 1.67* | [1.01–2.77] | 2.06* | [1.10–3.88] |
| Age at first sex > 17 | 1.16 | [0.84, 1.61] | 1.56 | [0.62, 3.90] | 1.62 | [0.91–2.86] | 0.92 | [0.43–1.97] |
| Partner 5 years older | 1.13 | [0.77, 1.67] | 2.56 | [0.77, 8.54] | 1.25 | [0.52–3.01] | 0.72 | [0.26–2.01] |
| Intended to get pregnant | 1.44** | [1.10, 1.89] | 0.89 | [0.41, 1.95] | 1.24 | [0.79–1.95] | 1.69 | [0.97–2.93] |
| Father has post primary education | 1.16 | [0.91, 1.47] | 0.99 | [0.53, 1.87] | | | | |
| Mother has post primary education | 1.1 | [0.86, 1.41] | | | 1.28 | [0.87–1.89] | | |
| Polygamous background | 1.02 | [0.82, 1.27] | 1.12 | [0.59, 2.15] | 1.68* | [1.10–2.57] | 1.48 | [0.85–2.59] |
| Either parent had child as teenager | 1.1 | [0.82, 1.47] | 0.42 | [0.16, 1.08] | 0.74 | [0.46–1.18] | 0.73 | [0.30–1.78] |
| Any sibling had child as teenager | 1.1 | [0.72, 1.68] | 1.55 | [0.34, 7.05] | 0.57 | [0.27–1.19] | 0.75 | [0.23–2.39] |
| Constant | 0.18** | [0.05, 0.62] | 0.03* | [0.00, 0.45] | 0.33 | [0.04–2.50] | 0.04** | [0.00–0.39] |

*** p<0.001

** p<0.01

* p<0.05

OR Odds Ratio, C.I Confidence Interval, RC Reference Category.

their respective pregnant teenage girls. Even within the Yoruba culture in the South-West Nigeria, the cultural narratives of *abiyamo* (motherhood) identifies the significant roles of mothers in supporting their offspring through childbearing [49].

Low attendance of antenatal in this study suggests the need for expansion of universal access to health coverage in terms of maternal and women's empowerment, as well as human capital to women in vulnerable group. In the same vein, the positive impact of education as enunciated in this study suggest that Nigerian education system needs to recognize both the direct and derived benefits of educating young women. The current policy across the country discriminates against pregnant young girls as they are either rusticated or voluntarily dropped out of school. There is the need to develop appropriate educational intervention for this demographic. There is also the need to address child marriage and adopt effective policies and strategies to reach married adolescents for improving empowerment and human capital of adolescent girls. Given the high rate of unplanned pregnancies in this study, interventions to prevent unintended pregnancies along with interventions to promote ANC utilization among women with unintended pregnancies should be prioritized. In addition, a focus on getting unmarried pregnant adolescents to utilize ANC services should be high on the agenda to improve the health of adolescent mothers and their babies. Finally, Alternative intervention to improve uptake of maternal health services among vulnerable adolescent childbearing mothers in Africa may have to consider the community mentor mothers' approach, either through enlisting older women in the community, or, through engagement of Community Health Extension Workers (CHEW). Addressing the maternal health situation in developing countries, and, in Africa especially, requires information on sub-group and subnational evidences [50–52]. This study is therefore significant in identifying vulnerable adolescent childbearing mothers as high-risk group in maternal health discourses in Africa and adding to literature on the influence of family context in the social determinants of health among this subgroup. The study concludes that identifying the role of parents and community in expanding access to ANC services among adolescent mothers is important in improving maternal health in developing countries.

## Limitations

The study is subject to limitations of a cross-sectional observational study with the potential for recall bias and lack of evidence of a temporal relationship. In addition, the study is not representative of the selected states, but rather, an adequate representation of their urban slum areas. We were unable to apply the DHS weight to the dataset because of the absence of a current sampling frame for those areas. However, we have an adequate sample size that is large enough and the estimate of the outcome variables in this study is similar to that obtained from the most recent DHS for each of the selected states.

## Supporting information

**S1 File.**
(DTA)

## Author Contributions

**Conceptualization:** Akanni Ibukun Akinyemi, Elizabeth Omoluabi.

**Formal analysis:** Akanni Ibukun Akinyemi, Temitope Peter Erinfolami, Samuel Olinapekun Adebayo, Reni Elewonbi.

**Methodology:** Akanni Ibukun Akinyemi.

**Supervision:** Iqbal Shah.

**Writing – original draft:** Akanni Ibukun Akinyemi, Temitope Peter Erinfolami.

**Writing – review & editing:** Temitope Peter Erinfolami, Samuel Olinapekun Adebayo, Iqbal Shah, Elizabeth Omoluabi.

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
