## [Decision Letter · Decision Letter 0]

26 Apr 2021

PONE-D-21-00752

Family Context and Individual Characteristics in Antenatal Care Utilization among adolescent childbearing mothers in Urban Slums in Nigeria.

PLOS ONE

Dear Dr. Akinyemi,

Thank you for submitting your manuscript to PLOS ONE. After careful consideration, we feel that it has merit but does not fully meet PLOS ONE’s publication criteria as it currently stands. Therefore, we invite you to submit a revised version of the manuscript that addresses the points raised during the review process.

Two expert reviewers provide you with detailed comments for improvement while agreeing that the article has potential.

In addition there are other reservations:

PLOS ONE endorses the strobe statement checklist for cross-sectional studies, https://www.strobe-statement.org/fileadmin/Strobe/uploads/checklists/STROBE_checklist_v4_cross-sectional.pdf. You can use it as a guide in your revision since many of the comments of the reviewers address issues from the checklist lacking in some aspects. Among these for instance, is the selection of participants (you mention "randomly", but this is vague: based on what? Did you have a listing? ...), how did you deal with missing data, participants (section 13), ...A second reservation is data availability. PLOS ONE policy is that authors are required to make all data underlying the findings described fully available, without restriction, and from the time of publication. PLOS allows rare exceptions to address legal and ethical concerns. See the PLOS Data Policy and FAQ for detailed information. This is not currently the case. It is wrongly stated that “All relevant data are within the manuscript and/or its Supporting Information files”. This is not true. Let me remind you the possibilities:

If the data are held or will be held in a public repository, include URLs, accession numbers or DOIs. If thisinformation will only be available after acceptance, indicate this by ticking the box below. For example: All XXX files are available from the XXX database (accession number(s) XXX, XXX.).If the micro data are all contained within the manuscript and/or Supporting Information files, enter the following: All relevant data are within the manuscript and its Supporting Information files.If neither of these applies but you are able to provide details of access elsewhere, with or without limitations, please do so.
Please also include the questionnaire used in the study.In table 2 you should indicate within the colums that the comparison is not "fair" in the sense that DHS refers to the whole state, as I understand, and not to the urban slums. You can be more explicit in the header, probably include Slums Adolescent Survey vs State-level DHS.I think the bivariate analysis is useful. Do not exclude it.Regarding the inclusion of 8 visits, you can include it in the analysis if you like, otherwise, admitting the change in recommendation from 4 to 8 visits, you can argue why it is still interesting to look at 4 visits.While you have to improve the description of the variables retained for analysis (items 7 and 8 of strobe statement), this is particularly important regarding the wealth variables, as manifested by the reviewers. Are you defining quintiles based on this questionnaire or are you using wealth cuts according to the DHS? That would make results more comparable. In particular, since only slum areas are included. Also social status is not described in enough detail to allow for replication. There could also be a high overlap between wealth and social status, in which case that could explain non-significance in the multivariate analysis and high significance in the bivariate analysis.Regarding the regression, you should include group tests of significance in table 4 and 5, since the tests for categorical variables that you report (individual) are less interesting since they compare to the referece category.Note also PLOS ONE guidelines for reporting statistical results, https://journals.plos.org/plosone/s/submission-guidelines#loc-statistical-reporting. In particular

**Regression analyses.** Include the full results of any regression analysis performed as a supplementary file. Include all estimated regression coefficients, their standard error, p-values, and confidence intervals, as well as the measures of goodness of fit.**Reporting parameters.** Test statistics (F/t/r) and associated degrees of freedom should be provided. Effect sizes and confidence intervals should be reported where appropriate. If percentages are provided, the numerator and denominator should also be given.**P-values. **Report exact p-values for all values greater than or equal to 0.001. P-values less than 0.001 may be expressed as p < 0.001, or as exponentials in studies of genetic associations.

We look forward to receiving your revised manuscript.

Kind regards,

José Antonio Ortega, Ph.D.

Academic Editor

PLOS ONE

Journal Requirements:

4. We note that Figure 1 in your submission contains map images which may be copyrighted.

We require you to either (a) present written permission from the copyright holder to publish this figure specifically under the CC BY 4.0 license, or (b) remove the figure from your submission:

b. If you are unable to obtain permission from the original copyright holder to publish this figure under the CC BY 4.0 license or if the copyright holder’s requirements are incompatible with the CC BY 4.0 license, please either i) remove the figure or ii) supply a replacement figure that complies with the CC BY 4.0 license. Please check copyright information on all replacement figures and update the figure caption with source information. If applicable, please specify in the figure caption text when a figure is similar but not identical to the original image and is therefore for illustrative purposes only.

6. Please include your tables as part of your main manuscript and remove the individual files. Please note that supplementary tables should be uploaded as separate "supporting information" files.

7. We note that Figure 1 also includes an image of individuals.

As per the PLOS ONE policy (http://journals.plos.org/plosone/s/submission-guidelines#loc-human-subjects-research) on papers that include identifying, or potentially identifying, information, the individual(s) or parent(s)/guardian(s) must be informed of the terms of the PLOS open-access (CC-BY) license and provide specific permission for publication of these details under the terms of this license.

Please download the Consent Form for Publication in a PLOS Journal (http://journals.plos.org/plosone/s/file?id=8ce6/plos-consent-form-english.pdf).

The signed consent form should not be submitted with the manuscript, but should be securely filed in the individual's case notes.

Please amend the methods section and ethics statement of the manuscript to explicitly state that the patient/participant has provided consent for publication: “The individual in this manuscript has given written informed consent (as outlined in PLOS consent form) to publish these case details”.

Reviewers' comments:

Reviewer's Responses to Questions

**Comments to the Author**

1. Is the manuscript technically sound, and do the data support the conclusions?

Reviewer #1: Yes

Reviewer #2: Yes

2. Has the statistical analysis been performed appropriately and rigorously? 

Reviewer #1: Yes

Reviewer #2: Yes

3. Have the authors made all data underlying the findings in their manuscript fully available?

Reviewer #1: Yes

Reviewer #2: Yes

4. Is the manuscript presented in an intelligible fashion and written in standard English?

Reviewer #1: Yes

Reviewer #2: No

5. Review Comments to the Author

Reviewer #1: The study investigates the role of family context and other determinants in adolescent mother access to antenatal care (ANC). Findings show, among other things, that having lost both parents negatively influences ANC access, while parents' education is not a relevant determinant.

The study is novel in its analysis of the family context and sheds a light in an often marginalized area. Findings on other factors are also in line with previous literature. The reviewer believes the paper would be a valid addition to PLOS ONE, although a few revisions are advised.

1) The Introduction section should provide more insights on the situation in Nigeria urban slums, considering the focus of the paper. Information on average ANC participation in these areas, or on adolescent health in general could prove useful for the reader.

2) The Measurement of Variables section should present a cleared explanation of how the social status indicator has been calculated, considering its relevance in the findings.

3) The role of wealth is not discussed in the Results/Discussion sections, although it is marginally significant. Results are also not reflective of previous literature on the role of wealth on ANC (i.e. the authors find that being wealthier is not correlated with a higher probability of attending 4ANC+). The authors should present hypotheses and potential explanations of their findings to explain why the situation in which households leave in the slums would reduce the importance of wealth in influencing access to ANC.

Reviewer #2: Reviewer feedback

BACKGROUND

Line 6-6: “The greatest burden is concentrated in West and Central Africa where 6 per-cent of adolescents reported births before age 15 and 28 per cent of women between the ages of 20 and 24 reported a birth before age 18.” The authors need to cite compared to what is the West and Central Africa have the greatest burden.

Line 26 -27: Research evidence from 26 developing countries consistently underscores the importance of ANC among childbearing 27 adolescents. What are some of the positive impacts of ANC for childbearing adolescents? For instance, what percent of death reduction is attributed to ANC?

Line 11-26: This section may need more argument why they have conducted their study. For instance, is there no study that examined the influence of family context in ANC uptake among childbearing adolescents in urban slums in Nigeria? If any, what are the methodological and knowledge gaps you wanted to fill on the existing evidence? This section may benefit from more professional and scientific critics of existing evidences to call out their paper’s strong side.

METHODS

Line 31-32: Were there any special ethical procedures followed for adolescents? Does Nigeria’s Research Ethics Guideline allow to take direct consent from women less than 18 years of age? Any assent or witness used during consent taking?

How was slum area defined in this paper? Did you take slum residents based on a set of criteria or slum sites/areas in general? Please provide brief explanation about the slum areas you studied.

Line 38-39: Why did you decide to include those having at least a child less than 5 years? Why not less than 1, why not less than 3 years? Brief explanation if you have any reason for your inclusion criteria.

Page 5: The data collection was implemented between July and October 2018. Why did it take this much time for data collection alone or does the period include the wider study period including analysis and report writing?

Ethical approval related information is repeated between pages 4 and 5.

Page 5: Measurement of variables: WHO has recommended at least 8 visits for a positive pregnancy experience? You have cited the World Health Organization (2016) minimum recommendations but the cut off you applied is older. The 2016 recommendation is a minimum of 8 contacts. Did you consider this or it is because Nigeria didn’t start implementing the new recommendation? Why did you decide a minimum of 4 visits rather than 8? Unless you justify this, you may need to re-analyze in the whole of the paper.

RESULT

Page 6, Line 17-25…the comparisons between the current study and DHS is not in the right place. Better move it to the Discussion part.

Line 29-38: I don’t see the importance of presenting this section (the Bivariate analysis). It is repeated under the Binary; logistic regression model except the later shows net effects…Just explain how you used the bivariate analysis to select the variables entered to the multivariate model and focus on the multivariate findings.

Page 7, Line 7-24: The interpretation approach is not inviting for readers. I counted that the word “Those” was repeated 11 times in this section and this indicates that the authors need to be careful in articulating their findings. It need to be re-written.

Page 7, Line 30-33: Authors have associated the influence of polygamous marriage in the North to 4+ANC visits. The possible explanation is not clear. The authors need to describe how ANC visit is associated with polygamous marriage and it has to be supported with evidence. Readers do not want to read the authors’ hypotheses but their evidence-based explanations. Plus, the “RESULT” section is not the right place to include possible explanations. The result section should only cover the findings in a simple language. Possible explanations and further interpretations of findings should be addressed under the Discussion part of the paper.

DISCUSSION

The first paragraph on page 9 (mental health) issue seems out of context. Mental health was never mentioned in the previous sections.

It would be advisable if the authors could come up with an overall comprehensive conclusion at the end of the discussion part. Recommendations are already mixed in the discussion.

6. PLOS authors have the option to publish the peer review history of their article (what does this mean?). If published, this will include your full peer review and any attached files.

Reviewer #1: No

Reviewer #2: No

---

## [Author Response · Author response to Decision Letter 0]

10 Nov 2021

Comment Reviewer 1 Response

The Introduction section should provide more insights on the situation in Nigeria urban slums, considering the focus of the paper. Information on average ANC participation in these areas, or on adolescent health in general could prove useful for the reader.

 The introduction is reviewed to reflect this. The last paragraph of the introduction already contextualize the problem in urban slums in Nigeria. However, there is no evidence on ANC participation in slum areas in Nigeria.

The Measurement of Variables section should present a clear explanation of how the social status indicator has been calculated, considering its relevance in the findings.

 reviewed 

The role of wealth is not discussed in the Results/Discussion sections, although it is marginally significant. Results are also not reflective of previous literature on the role of wealth on ANC (i.e., the authors find that being wealthier is not correlated with a higher probability of attending 4ANC+). The authors should present hypotheses and potential explanations of their findings to explain why the situation in which households leave in the slums would reduce the importance of wealth in influencing access to ANC.

 Reviewed to reflect this

Comment Reviewer 2 Response

BACKGROUND 

Line 6-6: “The greatest burden is concentrated in West and Central Africa where 6 per-cent of adolescents reported births before age 15 and 28 per cent of women between the ages of 20 and 24 reported a birth before age 18.” The authors need to cite compared to what is the West and Central Africa have the greatest burden.

 The sentence is now rephrased.

Line 26 -27: Research evidence from developing countries consistently underscores the importance of ANC among childbearing adolescents. What are some of the positive impacts of ANC for childbearing adolescents? For instance, what percent of death reduction is attributed to ANC?

 Reviewed 

Line 11-26: This section may need more argument why they have conducted their study. For instance, is there no study that examined the influence of family context in ANC uptake among childbearing adolescents in urban slums in Nigeria? If any, what are the methodological and knowledge gaps you wanted to fill on the existing evidence? This section may benefit from more professional and scientific critics of existing evidence to call out their paper’s strong side.

I did not find any study that has linked family context with ANC in Nigerian urban slums.

METHODS 

Line 31-32: Were there any special ethical procedures followed for adolescents? Does Nigeria’s Research Ethics Guideline allow to take direct consent from women less than 18 years of age? Any assent or witness used during consent taking?

 Reviewed

How was slum area defined in this paper? Did you take slum residents based on a set of criteria or slum sites/areas in general? Please provide brief explanation about the slum areas you studied.

 In line with UNHabitat definition

Line 38-39: Why did you decide to include those having at least a child less than 5 years? Why not less than 1, why not less than 3 years? Brief explanation if you have any reason for your inclusion criteria.

 Experience suggest that the sample frame be more open in order to get adequate sample.

Page 5: The data collection was implemented between July and October 2018. Why did it take this much time for data collection alone or does the period include the wider study period including analysis and report writing?

 Data was collected in 3 different States with a focus on each state per time. In Lagos and Oyo, data collection was between July 3 and August 6, while in Kaduna, data collection was between August 15 to October 9. In addition, data collection extended across all the locations because of callback to households where the selected persons were not immediately available for interviews

Ethical approval related information is repeated between pages 4 and 5.

 Reviewed

The duplication in the method section has been deleted

Page 5: Measurement of variables: WHO has recommended at least 8 visits for a positive pregnancy experience? You have cited the World Health Organization (2016) minimum recommendations but the cut off you applied is older. The 2016 recommendation is a minimum of 8 contacts. Did you consider this, or it is because Nigeria didn’t start implementing the new recommendation? Why did you decide a minimum of 4 visits rather than 8? Unless you justify this, you may need to re-analyze in the whole of the paper.

 Reviewed

RESULT 

Page 6, Line 17-25…the comparisons between the current study and DHS is not in the right place. Better move it to the Discussion part.

 DHS evidence deleted from the table

Line 29-38: I do not see the importance of presenting this section (the Bivariate analysis). It is repeated under the Binary; logistic regression model except the later shows net effects…Just explain how you used the bivariate analysis to select the variables entered to the multivariate model and focus on the multivariate findings.

 Reviewed 

Another reviewer recommended that the bivariate analyses are particularly important to the study and should be left as it is

Page 7, Line 7-24: The interpretation approach is not inviting for readers. I counted that the word “Those” was repeated 11 times in this section and this indicates that the authors need to be careful in articulating their findings. It needs to be re-written.

 Reviewed

Page 7, Line 30-33: Authors have associated the influence of polygamous marriage in the North to 4+ANC visits. The possible explanation is not clear. The authors need to describe how ANC visit is associated with polygamous marriage and it must be supported with evidence. Readers do not want to read the authors’ hypotheses but their evidence-based explanations. Plus, the “RESULT” section is not the right place to include possible explanations. The result section should only cover the findings in a simple language. Possible explanations and further interpretations of findings should be addressed under the Discussion part of the paper.

I did not find any study linking ANC to polygamous marriage

DISCUSSION 

The first paragraph on page 9 (mental health) issue seems out of context. Mental health was never mentioned in the previous sections.

 Reviewed

It would be advisable if the authors could come up with an overall comprehensive conclusion at the end of the discussion part. Recommendations are already mixed in the discussion.

 Reviewed and conclusion added

---

## [Editor Report · Decision Letter 1]

15 Nov 2021

Family Context and Individual Characteristics in Antenatal Care Utilization among adolescent childbearing mothers in Urban Slums in Nigeria.

PONE-D-21-00752R1

Dear Dr. Akinyemi,

We’re pleased to inform you that your manuscript has been judged scientifically suitable for publication and will be formally accepted for publication once it meets all outstanding technical requirements.

Kind regards,

José Antonio Ortega, Ph.D.

Academic Editor

PLOS ONE

Additional Editor Comments (optional):

The main issues raised in the discussion have been dealt with and the article fulfills PLOS ONE publication criteria in the opinion of the editor, congratulations. It's not been judged necessary to send back the article to the reviewers.
---

## [Editor Report · Acceptance letter]

17 Nov 2021

PONE-D-21-00752R1 

Family Context and Individual Characteristics in Antenatal Care Utilization among Adolescent Childbearing Mothers in Urban Slums in Nigeria. 

Dear Dr. Akinyemi:

I'm pleased to inform you that your manuscript has been deemed suitable for publication in PLOS ONE. Congratulations! Your manuscript is now with our production department. 

Kind regards, 

on behalf of

Dr. José Antonio Ortega 

Academic Editor

PLOS ONE